# Towards Precision Nutrition: A Novel Concept Linking Phytochemicals, Immune Response and Honey Bee Health

**DOI:** 10.3390/insects10110401

**Published:** 2019-11-12

**Authors:** Pedro Negri, Ethel Villalobos, Nicolás Szawarski, Natalia Damiani, Liesel Gende, Melisa Garrido, Matías Maggi, Silvina Quintana, Lorenzo Lamattina, Martin Eguaras

**Affiliations:** 1Centro de Investigación en Abejas Sociales (CIAS), Universidad Nacional de Mar del Plata (UNMdP), Deán Funes 3350, Mar del Plata CP 7600, Argentina; n.szawarski@gmail.com (N.S.); ndamiani@mdp.edu.ar (N.D.); lgende@mdp.edu.ar (L.G.); pmgarrid@mdp.edu.ar (M.G.); biomaggi@gmail.com (M.M.); biologiamolecular@farestaie.com.ar (S.Q.); mjeguaras@gmail.com (M.E.); 2Consejo Nacional de Investigaciones Científicas y Técnicas (CONICET), Godoy Cruz 2290, Buenos Aires C1425FQB, Argentina; lolama@mdp.edu.ar; 3Plant and Environmental Protection Sciences, College of Tropical Agriculture and Human Resources, University of Hawaii at Manoa, 3050 Maile Way, 310 Gilmore Hall, Honolulu, HI 96822, USA; emv@hawaii.edu; 4Instituto de Investigaciones Biológicas (IIB-CONICET), UNMdP, Dean Funes 3350, Mar del Plata CP 7600, Argentina

**Keywords:** *Apis mellifera*, nutrition, immunity, *Varroa*, cold stress, signaling pathways

## Abstract

The high annual losses of managed honey bees (*Apis mellifera*) has attracted intensive attention, and scientists have dedicated much effort trying to identify the stresses affecting bees. There are, however, no simple answers; rather, research suggests multifactorial effects. Several works have been reported highlighting the relationship between bees’ immunosuppression and the effects of malnutrition, parasites, pathogens, agrochemical and beekeeping pesticides exposure, forage dearth and cold stress. Here we analyze a possible connection between immunity-related signaling pathways that could be involved in the response to the stress resulted from *Varroa*-virus association and cold stress during winter. The analysis was made understanding the honey bee as a superorganism, where individuals are integrated and interacting within the colony, going from social to individual immune responses. We propose the term “Precision Nutrition” as a way to think and study bees’ nutrition in the search for key molecules which would be able to strengthen colonies’ responses to any or all of those stresses combined.

## 1. Introduction

Honey bees (*Apis mellifera*) are key pollinators, playing a vital role in ecosystem maintenance and the stability of crop yields [1]. Unfortunately, colony losses and colony depopulation of the European honey bee *A. mellifera* was reported for several years, being particularly high in the USA [2,3,4,5], where this phenomenon reaches levels up to 50% [6]. The losses of managed honey bees are of great concern, and this has attracted intensive attention, where scientists have dedicated much of their work to uncover the stresses affecting bees [4,7].

Deciphering the complex interactions between diseases, environmental factors and internal colony conditions, which can influence fitness and survival of honey bees, is very important.

Natural and/or anthropogenic stresses include exposure to agrochemicals and beekeeping pesticides, forage dearth, seasonal changes, environmental stress, among others [7,8,9]. Internal colony factors that are known to impact colony survival include parasites, pathogens, nutrition and genetics [8,9]. Diet and malnutrition are presently considered to be crucial, not only to individual health, but also to colony fitness [10]. Now, as a general consensus, it is considered that multifactorial elements contribute to the bees’ immunosuppression, leading to the weakening of the colonies and to the reported worldwide colony losses [7,10,11].

Based upon published research to date, it seems that the biggest challenge for honey bees derives from the immunosuppression produced by the combined effects of the Varroa mite (*Varroa destructor*), viruses, pesticides and malnutrition. At the same time, the compound effects of malnutrition, *Varroa* and viruses have been reported to have a greater impact during the winter season [12,13,14]. In addition to the forage dearth and host-parasite dynamics associated with winter, cold stress per se represents another threat to be overcome by weakened honey bee colonies [12,13,14,15,16,17].

Understanding the consequences of the factors detailed above should help to improve the management of honey bees under commercial beekeeping regimes. Below, we describe the scenario affecting *A. mellifera* with focus upon: Immunity, nutrition, the overwintering phase, *Varroa* parasitism and the main viruses vectored by the mite. Then, we explore a possible connection between the combination of different stresses and the main signaling pathways that could be involved in bees’ immune response.

Within the context described before, we will focus on the possible roles that could be played by two natural-occurring molecules: Abscisic acid (ABA) and nitric oxide (NO). We will explain how ABA could represent a key phytochemical, and NO a key signaling molecule involved in our honey bee’s immunity. Both molecules, ABA and NO, have been proven to be involved in honey bees’ immunity (see below for citations and details). At the same time, these two molecules are directly (ABA) or indirectly (NO, being synthesized from the essential amino acid L-arginine, see below) connected to a bee’s nutrition. Together, ABA and NO gather a substantial amount of evidence that could justify our analysis. This evidence was obtained at individual and colony level, using molecular, cellular and field approaches, among others (detailed below). However, we would like to highlight that the integrative analysis performed here should be taken just as an example, aiming to illustrate our vision.

In this review, we pretend to challenge our view regarding nutrition and bees’ strength. At the end, we propose a way to think and study about honey bees’ nutrition in the search for complementary practices to boost colonies’ fitness associated with beekeeping operations: “Precision Nutrition”.

## 2. Honey Bee Immunity

There is a major paradox in our understanding of honey bee immunity: As a social insect gathered within colonies, bees count on a particular level of immunity which is social immunity (see above). However, the high population density in a bee colony also implies a high rate of disease transmission among individuals, and this could represent a major challenge for individual immunity. Thus, the combined social and individual defense mechanisms of *A. mellifera* are necessary to confront any threat that could compromise the colony’s fitness [14,15,17].

### 2.1. Social Immunity

As eusocial insects, honeybee colonies form superorganisms, in which nest mates cooperate and use collective behaviors to combat parasites. Cremer et al. [18] introduces the term “social immunity” to describe the colony-level disease protection achieved through collective defenses of colony members. These defenses include behavioral, organizational and physiological components.

Social immunity behaviors are based on the ability of each individual bee to communicate and to respond to nest conditions, and to subsequently make individual choices that affect the collective superorganism.

The social immune processes deployed by honeybees include nest hygiene, secretion of antiseptic compounds to reduce or prevent disease, collection of plant-derived compounds that enhance colony health, thermoregulation changes associated with disease detection and control and defensive behaviors to protect the nest [19].

#### 2.1.1. Nest Hygiene

Nest hygiene [20,21] was first described in the context of American foulbrood disease, and includes several types of behavior, expressed by individual worker bees (females), which result in the removal of diseased and/or dead individuals from the hive space. In the case of the detection of a *Varroa*-infested pupa, the infested individual is removed, thus reducing or preventing the mite’s reproduction [22,23]. This last phenomena is a heritable trait, commonly known as *Varroa* sensitive hygiene (VSH), or suppressed mite reproduction (SMR) [24]. When worker bees perform hygienic behavior within a colony, this action confers colony-level resistance against bee brood diseases [25].

However, it is important to highlight that brood removal involves the sacrifice of the infested individual. Thus, the chemical or temperature cues utilized should be very accurate, and hypothetically, once the brood cells are capped, the detection of abnormalities in developing (pre)-pupae could be more challenging for bees.

Hygienic behavior can also include grooming behaviors among nest mates, which enables bees to remove ectoparasites (like *Varroa*), dust and pollen from their own bodies, and helps disperse pheromones [26]. Autogrooming stimulates the allogrooming, which starts when bees perceive another bee performing the grooming dance, and generally involves several nestmates acting collaboratively. Both autogrooming and allogrooming could be useful in detecting and removing parasites [27].

#### 2.1.2. Antiseptic Compounds

Honey bees secrete the antiseptic enzyme glucose oxidase (GOX) throughout their colonies’ brood food and honey reserves, providing social immunity to nest mates. GOX is produced in the hypopharyngeal glands, and catalyzes the oxidation of β-d-glucose to gluconic acid (C_6_H_12_O_7_) and hydrogen peroxide (H_2_O_2_). H_2_O_2_ acts an antiseptic, inhibiting pathogen growth in the larval food of honey bees [28]. Thus, the larval food is known to have a high antibiotic power, due to food gland secretions containing potent antimicrobial compounds [29,30].

#### 2.1.3. Plant-Derived Compounds for Nutrition and Defense

The foraging bees provide carbohydrates, protein, mineral elements, lipids and water to satisfy the nutritional requirements of the colony [31,32]. While collecting plant products (nectar, pollen and plant resins), these foragers will inevitably also gather associated secondary plant metabolites that would impact upon the individual bee and the colony health [33,34].

Social living allows bees to store food while providing them with an opportunity to selectively choose among the variety of the stored products to satisfy individual needs and/or the colony’s health. Finstrom and Spivak [35] show that chalkbrood-infested colonies increased propolis foraging, leading to a decrease in the infection intensities. Results of Borba and Spivak [36] indicate that the propolis envelope serves as an antimicrobial layer around the colony, helping bees to protect the brood from *Paenibacillus larvae* infection. Gherman et al. [37] demonstrates that workers infected with the microsporidium parasite *Nosema* spp. prefer to consume honeys with a higher antibiotic activity, leading to a reduction of the microsporidian infection.

To survive seasonal changes in forage quality and availability, honey bees store resources within the hive, and complex nutrients within the bodies of long-lived workers [38,39]. These “diutinus” (long lasting) bees represent a good example of how individual endocrine signaling defines social behavior, being modulated by the environment and as one adaptation of bee colonies to temperate climates [40]. During winter, this diutinus phenotype achieves long life span, but they are not in diapause like other species of insects [40]. This superorganism-level nutritional adaptation is characterized by a high rate of the expression of vitellogenin (Vg) during the months leading up to winter [41,42,43]. The produced Vg plays important roles related to the tolerance to starvation, immunological function, oxidative stress, regulation of worker ontogeny and life span [38,44]. However, other stress conditions could alter the expression patterns of Vg. For example, the disease caused by *Nosema ceranae* reduces the transcripts of Vg [45,46], which in turn could lead to a reduction of its antioxidant capacity and immune priming.

#### 2.1.4. Thermoregulation

Honey bees are able regulate their nest temperatures with high precision, maintaining the right temperatures inside their nests to support brood development [47]. Besides, thermoregulation can also be part of the defense against pathogens through what is known as “behavioral fever”. Starks et al. [48] report an increase in brood temperature associated with the occurrence of *Ascosphaera apis*, a heat-sensitive fungal pathogen that causes the disease commonly known as chalk brood.

Among all other honey bee stresses, temperature is one ecological factor that affects honey bee survival [17]. In temperate climates, honey bees survive throughout winter by entering a distinct physiological and behavioral state [49]. Honey bee colonies spend much energy to maintain brood nest temperature in the range of 32–36 °C [17,47]. Indeed, the brood reared at temperatures below 32 °C shows delayed development, increased mortality, abnormal nervous system development and poor behavioral performance as adult bees [50,51].

Despite all the honey bee studies related to the overwintering period, we know little about the mechanisms related with the effects of cold temperatures per se at physiological level [13,14,17]. Understanding how low-temperature stress per se affects honey bee physiology and longevity could help to shape management strategies to improve overwintering success [17]. Yet, the impact of low temperature stress on the mortality of brood has not been systematically investigated [52].

#### 2.1.5. Nest Defense

Adult bees react against predators and conspecific thieves, defending the nest collectively, and depending on the subspecies, more or less aggressively. This defensive behavior is primarily defined by genetics, but it is also socially modulated by pheromone cues and occurs in response to ecological conditions [53,54]. Some reports suggest that high aggression could be associated with lower parasitization by *V. destructor*, and that low aggression could be linked to increased pesticide susceptibility [55,56]. Consequently, in high aggression colonies, the spread of the mites could be regulated according to the variation in adult grooming or hygienic behaviors, which results in reduced mite reproductive success [57]. Moreover, more aggressive colonies may have better food resources because they tend to forage at higher rates [58].

### 2.2. Individual Immunity: A Brief Summary

At the individual level, honey bees have several lines of innate immune defense against parasites and pathogens [59,60]. Physical and chemical barriers, including the exoskeleton cuticle and the peritrophic membranes lining the digestive tract, are a first line of defense that prevent invaders from adhering to or entering the body [10]. If a parasite or a pathogen breaches the physical and chemical barriers, honey bees can protect themselves with cellular and humoral immune responses which represent a second line of defense [10].

Humoral defenses refer to soluble effector molecules, such as antimicrobial peptides, complement-like proteins and enzymatic cascades that regulate melanin formation and clotting. Cellular immunity is comprised by cell-mediated responses like phagocytosis, nodulation, encapsulation and wound closure [10].

Innate immune responses are activated and/or modulated through the action of the highly-conserved pathways Toll, immune deficiency (Imd), Janus kinases/signal transducer and activator of transcription proteins (JAK/STAT) and c-Jun N-terminal kinase (JNK) [59,60]. First, bees were predicted to express only two-thirds of the immunity genes expressed in solitary insects, e.g., mosquito or fruit fly [59]. From this, it was suggested that the immune response in bees should rely on social immunity, while some specific immune factors are upregulated in response to infection [59]. Later, Barribeau et al. [60] found similarities in the immunity studied across a gradient of sociality in insects, suggesting that the reduced immune repertoire predates the evolution of sociality in bees. The authors proposed that those differences regarding the quantity of the expressed immunity genes might be the result of selection ruled by divergent pressures exerted by parasites and pathogens within the context of eusociality.

One of the most important immune responses (generally, but also for the present work) reported to date to play a key part in insects’ immunity is the one based on the action of phenoloxidase (PO) [61,62]. PO is involved in the formation of melanin, which produces pigmentation, but it can also be employed in the immune response to confront pathogens and parasites [63]. Upon injury, coagulation and melanization are characteristic responses integrated in the wound-healing process in insects [63]. The lower activity of PO during the early stages of larvae development seems to be correlated with the susceptibility to infection [61]. The relevance of PO activation in melanin synthesis, and thus in *A. mellifera* immunity, has been related with the bees’ ontogeny [64,65,66]. Indeed, melanization has been proposed to be a major immune response in adult bees through the evaluation of the enzymes’ activity [64,65,66]. It is important to note that this PO response comes at a high energetic cost to the individual insect; the enzyme’s main activating system (pro-PO) depends upon tyrosine, which derives from phenylalanine, a compound that can only be obtained from ingested food. In addition, the resulting defensive compounds that are produced are rich in nitrogen, which requires a resource investment on the part of the insect [63].

The honey bees’ antioxidant enzymes are of particular interest too, because these are those responsible for the detoxification of reactive oxygen species (ROS). All aerobic organisms generate ROS in the process of their oxidative metabolism [67,68]. These reactive oxygen species include the superoxide anion (O_2_^· –^), the hydroperoxyl radical (HO_2_^·^), hydrogen peroxide (H_2_O_2_) and the hydroxyl radical (·OH) among others. The ROS can cause the oxidation of proteins, RNAs and DNAs, and the peroxidation of membrane lipids. Imbalances between the production of free radicals and the generation of antioxidants (to detoxify the reactive intermediates or to repair the resulting damage) causes oxidative stress in living cells [67]. Both, Superoxide dismutases (SODs) and catalase (CAT) are the first lines of defense against oxygen free radicals; also others like glutathione S-transferase, glutathione peroxidase and glutathione reductase, all of which have been reported to occur in insects. Interestingly, the melanogenic processes (described above) contribute to the formation of cytotoxic molecules that have the capacity to interact with ROS and reactive nitrogen species (RNS) to provide an effective immune response for insects [62]. Indeed, ROS and RNS generated during melanogenesis have been implicated in the killing of parasites by insects [62].

Nitric oxide (NO) is a highly reactive RNS, being an unstable free radical gas that is produced by the oxidation of L-arginine to citrulline mediated by the enzyme NO synthase (NOS) [69]. Due to its nature, NO could either act as a second messenger within each cell, or work as a signal between contiguous cells [70]. Indeed, it has been proposed that NO could be a key molecule of invertebrate’s immune responses to confront parasites [70]. In fact, NO has been reported to exert a potent bactericidal effect, as well as a key signaling role triggered upon gut infection in mosquitoes and the common fruit fly (*Drosophila melanogaster*) [70,71,72]. In honey bees, it has been documented that NO acts as a signaling molecule during the first steps of hemocyte activation after non-self recognition, in wound healing/encapsulation and in response to lipopolysaccharide injection (LPS) [73,74,75].

Although a considerable amount of information is available about humoral immune reactions (represented mainly by antimicrobial-peptides-based defenses [10]), less is known about the cellular immune system in honey bees. The cellular immune response comprises wound healing, phagocytosis, virus killing, nodulation and the encapsulation of the intruder. All of these reactions are mediated by the insect blood cells, the hemocytes [76], and new published results reinforce the relevance of the hemocytes in honey bees’ immune defenses [77,78,79,80]. Relevant studies were performed in the last years regarding *A. mellifera* hemocytes’ characterization [81,82]. A deeper analysis about the relevance of studying honey bees´ immunity, integrating cellular and humoral responses, is presented below in this review.

### 2.3. Interaction Between Social and Individual Immunity

The paradox of honey bee social immunity is that individual bee immunity does matter. In a nest environment where everything is shared, the potential risks of infection are high. However, there are also advantages to the superorganism lifestyle; in particular, the ability of a colony to tap into diverse individual genetics, behavioral responses and environmental resources to enhance communal health.

One of the key components of eusociality is the overlap between generations, and in the case of honey bees there is extensive and intimate contact between adult and developing bees. Social behavior in the colony leads to nurturing behavior towards larvae, and in contrast, social immunity to control the spread of disease. Through the active removal of a sick brood, honey bees face the dangerous trade-off between the sacrifice of parasitized individuals and the growth of the colony as a whole: Hygienic behavior itself may also break down through *Varroa*-transmitted virus epidemics, placing limits upon its ability to control mite infestation [15,83]. Avoiding this breakdown is therefore critical to colony survival [15,83].

Interestingly, reported evidences show that immune-challenged bees with injected bacteria receive more allogrooming assistance than non-injected ones [84,85]. This suggests that bees can detect individuals that have an activated immune system [19], and possibly, that an enhanced individual immune response could trigger a more efficient social defense. This ability to respond to the health status of an individual of the colony may boost the overall immune response as a superorganism. The phenomena described above is also supported by the findings of Harpur et al. [86], describing the absence of genetic trade-offs between hygienic behavior and innate immunity in honey bees. These evidences suggest that artificial selection to increase hygienic behavior in honey bee colonies should not be expected to consequently compromise the individual innate immunity of such bees.

The high frequency of interactions and contact of adult nurse bees with developing larvae within uncapped cells provides an ample opportunity to develop and utilize social immunity mechanisms. In contrast, the capped brood phase presents a very different scenario, where an age group is almost physically isolated from the rest of the colony (Figure 1). Capped brood phase then acquires special significance within the epidemiological structure of the colony, as it represents a potential haven for pathogens and parasites seeking to avoid detection and removal. Pathogens may indeed adjust their virulence, either to escape detection during the open brood period, or to ensure the completion of the capped brood phase and the emergence of the (infected) young adult. Consequently, the virulence of brood pathogens at an individual level is often inversely correlated with the virulence of the pathogen at the colony level [15]. An interesting example of this is represented by the variability in the pathogenicity of the bacterial disease American foulbrood, related to the genotype of the strain: Members of ERIC II–IV were found to be highly virulent for individual larvae, which died in a short time. In contrast, larvae infected with the genotype ERIC I survived up to 12 days; hence, this strain was considered less virulent than ERIC II–IV for this life stage. Then, the opposite was found in the colony, where a higher virulence at the larval level was related to a lower virulence at the colony level [87,88].

All in all, the revelations stated before highlight the importance of studying individual immunity in bee brood to complement the latest advances regarding individual and social defenses in *A. mellifera*.

## 3. Nutrition and Immunity

### 3.1. Generalities

The connection between nutrition and immunity has been demonstrated in numerous organisms, and honey bees are not the exception. The study of nutrition’s impact is one of the most rapidly expanding research areas in bee biology, largely due to the association of colony losses with malnutrition and accompanying pathologies [9,10,32]. The combination of new molecular techniques with the availability of the honey bee genome are enhancing the power of our interpretation regarding the role of nutrition in honey bee health [10].

During the early stages of honey bees’ development, royal jelly makes up a great part of the larvae’s diet, while its quality strongly influences the in vitro rearing success as well [89]. The presence of the antibacterial substance 10-hydroxy-2-decenoic acid (10-HDA) suggests that royal jelly could play a significant anti-pathogenic role in the mid-guts of bee larvae [90]. Besides, within royal jelly, nurse bees secrete large amounts of unique major royal jelly proteins (MRJPs) [91]. These proteins are synthesized endogenously in the hypopharyngeal glands located in the head of the honey bee [91]. The functions of MRJPs go from nutritional contributions [92], to antibacterial activity protecting bee larvae [93] and participation as structural proteins increasing food jelly viscosity [94].

Honey bees find dietary protein basically in pollen, which contributes to the immune response by providing the essential amino acids needed for the synthesis of peptides [10]. A bee’s immunity is also boosted by carbohydrates from nectar and honey that provide energy for metabolic processes associated with both innate humoral and cellular reactions [10]. Thus, the impact of abundant and diverse floral plant sources among different bee species has been extensively studied [95,96].

In 2010, Alaux et al. [97] tested whether dietary protein quantity (represented by monofloral pollen) and diet diversity (presented as polyfloral pollen) could shape the baseline immuno-competence (IC) of *A. mellifera*. IC was evaluated by measuring the parameters of individual immunity, like hemocyte concentration and phenoloxidase activity (among others), and GOX activity as a parameter of social immunity. The results of this work suggest that there is a link between protein nutrition and immunity in honeybees, highlighting the critical role of resource availability on pollinator health. Recently, within the context of honey bees’ nutrition and floral resources, Ricigliano and co-workers [39] study the effects (pre and post-winter) of the forage environment in apiaries close to agricultural or non-agricultural landscapes (Conservation Research Program (CRP) lands) over the bees’ colonies strength. They find that the performance of honey bee colonies (adult bee mass and brood) and biomarkers associated with adequate nutrition (e.g., vitellogenin) are positively influenced by their foraging proximity to CRP lands. In straight relationship with the work described above, Branchiccela et al. [98] recently demonstrated that the nutritional state is related to the variety of pollen in the bees’ diet, and has a severe impact on bee colony strength and health, with both short and long-term consequences.

In 2016, Di Pasquale et al. [99] reported that changes in bee health were not connected to variations in pollen diversity, but rather to variations in pollen depletion and the quality related with an intensive agricultural system. The authors suggest that, even though pollen can be available in great quantities during the mass-flowering of some crops, it could fail to provide bees with a diet adequate for their development. Interestingly, Kriesell et al. [100] find that various bumblebee species visit different floral spectra for pollen collection, but nevertheless have highly similar pollen amino acid profiles. This suggests that, despite the different pollen foraging patterns, distinct bumblebee species share similarities regarding the requirement of specific amino acids in their diet. All in all, the findings cited above highlight the relevance of studying more about the precise nutritional requirements of most bee species.

The evidences described in the paragraphs above is integrated and illustrated in Figure 1.

A diverse set of effects of carbohydrate and protein sources over the transcriptional profiles of bees is evidenced lately through genomic approaches. It is reported that constituents found in honey up-regulate the detoxification pathways in the gut and genes associated with protein metabolism and oxidative reduction [101]. Those effects are not found to be induced by a sucrose solution or high fructose corn syrup, commonly used to feed bees in managed colonies [102]. Besides, pollen activates nutrient-sensing and metabolic pathways, and influences the expression of genes involved in longevity, immune function, the production of certain antimicrobial peptides and pesticide detoxification [42,103]. At the same time, it is important to take into consideration that protein sources based on pollen mixes could be contaminated with different pesticides [98].

Also, it is described that resins collected by bees affect the expression of immunity genes [104], and that propolis extracts reduce the toxicity of mycotoxins [105]. Indeed, it is demonstrated that nectar, pollen and resins contain secondary plant metabolites that have antimicrobial properties [34,106] and key molecules involved in *A. mellifera* immune response [101,107,108]. The harvesting, storage, and/or ingestion of non-nutritive compounds and plant materials are part of a self-medication strategy employed by social bees [19,35]. A review by Erler and Moritz (2016) distinguishes between defense compounds directly related to the bees’ diet, that is, pharmacophagy, which focuses upon the health-related benefits of the nectar and pollen consumed by the bees, and pharmacophory, which focuses on the role of the chemicals collected, but not eaten, such as resins that add protection to all the inhabitants of the colony [34].

Up to this point and as a general proposal, we would like to summarize that: (a) Honey is basically made from the flowers’ nectar (and also honeydew) collected by forager bees and (b) Honey contains pollen also. Thus, it could be hypothesized that: (a) Plant-derived compounds present in pollen and nectar should end up in honey and (b) the concentration of those compounds could change throughout the process, leading to honey as the final product. By this mean, honey could be interpreted as a potential pre-winter storage for self-medication throughout the season of forage dearth. With this, we also want to suggest that the concentration of phytochemicals should be studied also in honeys from different origins and regions.

### 3.2. Particularities: Phytochemicals as Dietary Treatments for Medication

In 2013, Mao et al. [101] determined that p-Coumaric acid (CouA), a phytochemical found in pollen and honey, up-regulates genes involved in detoxification processes as well as select antimicrobial peptides’ synthesis. Interestingly, when CouA was added to the diet, the metabolism of the acaricide coumaphos in the bees’ mid-gut increased significantly (Table 1). In 2017, Liao et al. [109] conducted a series of experiments in which they test if CouA or quercetin (Quer) enhances longevity and pesticide tolerance. Both dietary phytochemicals are associated with extended lifespan. They also showed that Quer enhanced the tolerance to two pyrethroids, while CouA follows a similar trend, but of a reduced magnitude (Table 1). Then Wong et al. [110] report that CouA and Quer, incorporated in the diet, enhances the survival of honey bees exposed to imidacloprid (IMI) at low concentrations. However, they find that both molecules have a negative effect at higher concentrations individually in chronic toxicity bioassays. Working with combinations of different concentrations of those phytochemicals, the authors find a biphasic concentration-dependent response of the honey bees’ IMI. These authors conclude that the protective effects of these plant-derived compounds against neonicotinoids effects over bees are limited based on their own inherent toxicity (Table 1).

In 2014, Strachecka et al. [113] reported that honey bees that consume caffeine (Caff) live longer, and were not infested with *Nosema* spp. Caff-treated honey bee workers have higher protein concentrations, and show increased activities of physiologically-relevant enzymes like the ones involved in the antioxidant response (Table 1). Recently, Bernklau et al. [114] studied the effects of Caff, gallic acid (GA), kaempferol (Kaemp) and CouA on survival and pathogen tolerance in honey bees. They show that bees supplemented with dietary phytochemicals survive longer and this enables bees to combat infection with *N. ceranae* by reducing spore-loads (Table 1). In addition, the relevance of incorporating GA into dietary supplements combined with nanotechnology to counteract *P. larvae* infection in honey bees is reported recently [118].

In 2015 it was reported that bumble bees infected with the intestinal parasite *Crithidia bombi* preferably choose to consume artificial nectar that contains nicotine (Nic), which at the same time reduces their parasite load [115] (Table 1). Then, Thorburn et al. [116] tested for interactions between the effects of Nic and anabasine (Ana) (found in the nectar of *Nicotiana* spp. plants) on the *C. bombi* load and the mortality in bumble bees (Table 1.). They find that when the experimental environmental conditions setup is variable, each alkaloid alone significantly decreases parasite loads. Interestingly, this effect is not observed when the alkaloids are combined, suggesting an antagonistic interaction. When the experiments are performed in stable environmental conditions, Nic significantly increases parasite loads, the opposite of its effect in the variable setup. In stable conditions, the authors find a positive relationship between Ana and parasite loads. Interestingly, these authors suggest an interesting interaction between phytochemicals, parasites and environmental variables, and they evidence that plant-derived compounds could be either toxic or medicinal, depending on context. This last work is very relevant for the purpose of the present review, highlighting the relevance of the effect of environmental conditions over the bees’ immunity and response to phytochemicals.

More recently, Palmer-Young et al. [112] examined the effects of amygdalin (Amyg), Ana, aucubin (Auc), catalpol (Catal), clove oil (Cloil), fumagillin (Fuma), Nic and thymol (Thym), after feeding honey bees with those phytochemicals (Table 1). They evaluate whether phytochemical consumption would counteract preexisting infection in mature bees, or mitigate infection in young bees. Generally, phytochemicals are well-tolerated at levels documented in nectar, honey and pollen, with the exception of the Cloil and Thym that increases mortality at high doses. They find that short-term phytochemical consumption reduces levels of deformed wing virus (DWV), significantly in young bees that are released into field colonies. However, the non-toxic doses of the phytochemicals evaluated do not alter infection with *Lotmaria passim* or *N. ceranae*. With the exception of Amyg, all the tested phytochemicals significantly increase the antimicrobial peptide hymenoptaecin expression in older bees after long-term consumption. Interestingly these authors describe that phytochemicals lack antiviral effects for pollen-deprived bees reared outside the colony.

During the past years it is reported that abscisic acid (ABA), a natural component present in nectar, honey and pollen [119,120], as well in honey bees [107,119], plays an important role in bees’ health. In 2015, Negri et al. [108] evaluated the effects of ABA on the bees’ immune responses by analyzing the effects of this molecule over the performance of small *A. mellifera* colonies throughout the winter season (Figure 1) (Table 1). They observe that ABA has an important effect at the individual level, stimulating the cellular and humoral innate responses, and at a colony level, where populations of adult bees supplemented with ABA are significantly bigger than control populations after winter. Then, in 2017, Negri et al. [75] reported new evidences regarding ABA and the activation of the cellular immune activation response in response to a bacterial elicitor (Table 1). At the same time, Ramirez et al. [52] showed that supplementing bees’ diets with ABA prevents low survival rate and accelerated adult emergence of in vitro reared honey bee larvae exposed to suboptimal temperatures below 32 °C (25 °C). In this work, the authors also show that ABA enhances the expression of genes involved in metabolic and stress responses (Table 1). Using the same experimental approach, Negri et al. [109] study the relationship between cold exposure, dietary ABA supplementation and the expression of genes involved in the immune response.

They find that low temperatures and ABA induce the expression of several immune-associated genes in honey bee larvae, supporting that the immune system is active during cold stress response and reinforcing the connection between honey bees’ response to cold stress and ABA (Table 1). Recently, Szawarski et al. [117] evaluated the effects of ABA in combination with two different beekeeping nutritional strategies to confront overwintering, one based on honey and the other on syrup supplementation. The results indicate that the ABA supplementation has positive effects on the population dynamics of the *A. mellifera* colonies during overwintering and decreased *Nosema* loads at colony level (prevalence) in both the nutritional strategies evaluated (Table 1).

Below we show an example of the kind of integrative analysis that could lead to the discovery of key phytochemicals that could be included into the bees’ diet to enhance their overall performance. We will focus on ABA as the phytochemical involved in the analysis. The reasons to focus on ABA are summarized in Table 1. Through a series of correlative papers, ABA gathers the greatest amount of evidence we could find in comparison with the other most cited phytochemicals at the moment (CouA, Quer, Nic, Ana and Caff). Abscisic acid is proven to be related to bee nutrition and immunity through different experimental approaches, at different levels of organization and in response to different kinds of threats (Table 1). The other phytochemicals found in the Table 1 represent also a great field for future research, and with great potentiality. However, more studies have to be performed to understand more about the effects of ABA, CouA, Quer, Nic, Ana and Caff over bees’ physiology at individual and colony level, integrating individual and social immunity with both biotic and abiotic stressors.

## 4. Honey Bees Within the Context of Innate Immunity Research

The insects’ innate immune system involves a diverse set of responses, including the production and secretion of antimicrobial peptides, phagocytosis and the degradation of pathogens, melanization and encapsulation [121]. Indeed, insect immune pathways share specific orthologous components with the innate immune system of vertebrates [122]. This suggests a shared root for the immune pathways and a selection to conserve many components over hundreds of millions of years [59,60]. What is descripted above highlights the relevance of performing integrative analyses, combining what has been reported in different species within the context of innate immunity, when looking for traces to understand a particular response in our model of study. In this case, the model of study is the honey bee, and we should put it into the context of what has been reported about insects’ and vertebrates’ innate immunity responses.

The global distribution of honey bees has resulted in an increased exposure of this species to pathogens and parasites from diverse origins. As such, this insect has now been added to the list of biological models used to study innate, immune non-self recognition [77]. Based primarily upon extensive searches for orthologs of well-studied insects including fruit flies, mosquitoes and moth species, Evans et al. [59] propose honey bee models for four non-autonomous pathways implicated in inducible host defense, i.e., Toll, Imd, JAK/STAT and JNK. While these pathways engage in cross-talk and can direct some of the same immune effectors, they have well-defined structures and interaction sets, and are best tackled as individual entities [59].

Below, we will focus on the JAK/STAT pathway and its relationship with NO and ABA. Then, the relevance of this analysis will be revealed within the context of *V. destructor* parasitism and winter (where the Toll pathway will play a key part, as well) (Figure 1).

### 4.1. Janus Kinases/signal Transducer and Activator of Transcription Proteins (JAK/STAT) and Nitric Oxide (NO) Signaling

The JAK-STAT pathway, initially characterized for its role in development and hemocyte proliferation, is shown to also respond to bacterial and viral infections [123]. The family of transcription factors’ signal transducers and activators of transcription (STAT) activates the expression of immune genes in response to cytokine signaling [124]. Indeed, cytokines such as interleukins and interferons play a central role in regulating and coordinating the immune response through this pathway [125].

This pathway, which allows organisms to respond to extracellular signals, appears to be shared among many different groups of animals; from the cytokine-driven regulation of innate and acquired immunity responses in mammals [125], to the immune responses in insects.

Bioinformatic analysis reveals the presence of the cytokine receptor homolog *Domeless*, and four complement-like thioester-containing proteins (TEPs) indicate that this mechanism may be common across insects, and is intact in honey bees as well as in flies [59]. Thus, cytokine-like molecules should play a key role in an insect’s immune response through the action of the JAK/STAT pathway. The relevance of cytokine-like molecules and JAK/STAT activation shall be highlighted below in this work.

For the purpose of this work, it results relevant that a member of the STAT family has been reported to regulate the transcriptional activation of nitric oxide synthase (NOS, the enzyme responsible for nitric oxide production [69]) in response to interferon-gamma (IFN-γ) [124]. As it was mentioned before, NO is produced by the action of NOS enzymes [69] and crosses cellular membranes with ease, enabling very efficient responses by transmitting both intra- and intercellular signals [69,70]. In addition, members of the families of transcription factors Rel/NF-κB and JAK/STAT, which play central roles in the cytokine-driven regulation of both innate and acquired immunity in vertebrates, are activated by NO [126].

Besides playing the role of a signaling molecule, when produced in big amounts, NO is toxic to many kinds of pathogens, including viruses, fungi, bacteria and parasites; the latter include intracellular and extracellular invaders [70]. Interestingly, after being produced by activated immune cells, NO can diffuse from where it was synthesized, and can initiate cytotoxic reactions at distant sites, either by reacting per se, or together with other molecules [69], including melanin [62]. The relevance of the participation of NO in this kind of response in honey bees’ immunity will be highlighted below in this work.

In vertebrates, two main types of NOS enzymes are found: Constitutive (cNOS) and inducible (iNOS). The cNOS are part of the basal metabolism of cells, and are rapidly activated through changes in intracellular calcium levels [70]. By contrast, the iNOS isoform is absent in non-activated cells, but is rapidly synthesized in response to the pro-inflammatory cytokines. Inducible NOS catalyzes NO synthesis until the substrate is depleted, being able to produce up to 1000 times more NO than do the constitutive enzymes [70]. It is established that NOS enzymes found in invertebrates function either as an inducible or as a constitutive form depending on the insult (reviewed in [70]). This last means that insects’ NOS can increase the levels of NO by having its expression up-regulated or by being activated. Although the presence of a single *NOS* gene is reported within the *A. mellifera* genome (*AmNOS*, AB204558) more efforts should be done to characterize this enzyme.

To achieve pleiotropic effects, NOS induction is critical in responses under the control of JAK/STAT signaling [127,128]. IFN-γ interacts with its membrane receptors and activates a Janus kinase (JAK), which in turn activates STAT-1 by phosphorylating specific residues [124]. Indeed, IFN-γ is well known to play crucial roles in several aspects of the immune response, like the activation of the *iNOS* gene expression [127,128].

The JAK/STAT pathway is also shown to participate in an antiviral response in *Drosophila* [129]. In the mosquito *Anopheles gambiae* the JAK/STAT pathway is shown to be activated in response to bacterial challenge [130]. Indeed, *Ag*STAT-A mediates the transcriptional activation of *NOS* in response to infection [124]. These findings provide direct evidence that, in insects, *NOS* expression is also regulated by the STAT pathway, and this suggests that the organization of this signaling cascade precedes the divergence of insects and vertebrates [124]. Recently, it was demonstrated that ABA supplementation of a *Plasmodium falciparum*-infected blood meal increased expression of a mosquito’s *NOS* and reduced infection prevalence in a NO-dependent manner [131]. This last evidence represents that: a) ABA could be considered as a medicinal phytochemical and/or play an active role within the immune response of another insect model besides bees, and b) that JAK/STAT, NO and ABA signaling could be related.

### 4.2. Abscisic acid in Animals’ Immunity

Historically, ABA is studied as a phytohormone involved in the fundamental physiological processes of higher plants, including responses to abiotic stresses (temperature, light, drought) [132] and to pathogens [133], the regulation of pollen germination [134], the regulation of seed dormancy and germination [132] and the control of stomatal closure [132]. As mentioned above in this review, the presence of ABA is demonstrated in nectar and honey [119,120,135,136], as well as in honey bees [52,107,119,120].

In animals, ABA is considered as a pro-inflammatory cytokine involved in key processes leading to the activation of innate immune responses like phagocytosis, reactive oxygen species and NO production, and the chemotaxis of human granulocytes [137]. Through a series of correlative works, it is reported that ABA plays a role in immune and anti-stress responses in honey bees [52,75,107,109,117]. Here we would like to highlight the relevance that studying, connecting and integrating the evidences found in both vertebrates and invertebrates might have to understand the effects of plant-derived compounds over animals’ innate immunity.

Below we integrate responses to diverse stresses, considering the possible relation between ABA acting as a cytokine in animals, the JAK/STAT pathway (activated by cytokines, see above) and NO.

### 4.3. JAK/STAT and Cold Stress

Evidence of immune response enhancement after cold exposure suggests that cold activates the insects’ immune system, particularly trough the JAK/STAT pathway [138,139,140,141,142]. In flies, the extracellular glycosylated protein unpaired (Upd) acts as a ligand that activates the JAK/STAT pathway and regulates the expression of members of the TEP (see above for reference to TEPs) and TOT families (*TotA*, *TotC* and *TotM*), which in turn promotes the phagocytic activity of hemocytes in response to bacterial challenge [125]. Zhang et al. [138] find that *TotA*, *TotC* and *TotM* were up-regulated after cold exposures in *D. melanogaster*. Vermeulen et al. [140] report that cold stress induces the up-regulation of several *TEPs*, and considers that the activation of the immune system may be part of a specific aspect of the cold stress response rather than a general stress response. Another work finds that the expression of the genes of this family could be induced in *D. melanogaster* by varied stressors, including heat shock, cold shock, septic injury or bacterial infection [141]. More recently, Salehipourshirazi [141] found that acute cold exposure increased hemocyte concentration and wound-induced melanization and triggered the up-regulation of the JAK/STAT pathway. Working with another insect model, Krams et al. [139] find that an enhanced encapsulation response (which is a JAK/STAT-driven response [125]) is associated with higher winter survival in water striders.

### 4.4. JAK/STAT and Wound-Healing

The JAK-STAT pathway is reported to participate in multicellular-humoral responses in insects, where the co-participation of hemocytes with the melanization and coagulation responses are essential, like wound-healing and encapsulation [125]. Indeed, the JAK/STAT pathway plays a key role during the response of *Drosophila* to the invasion of parasitic wasps [125], which are known to introduce viruses and anticoagulant molecules into the hemolymph to counteract the immune defenses of the host [143]. Such defenses include hemocyte proliferation and differentiation, and the activation of the phenoloxidase cascade leading to melanization, all of them being signaled by the JAK/STAT pathway [125]. As could be expected, the wound-healing response of honey bees is also impaired by *V. destructor* [143,144]. Indeed, the relative expression of pro-phenoloxidase gen (*proPO)* as well as PO activity, decreases in *Varroa*-infested individuals [144,145]. This last indicates that *V. destructor* significantly influences the phenoloxidase-dependent response of melanization in honey bees, which is a multicellular-humoral defense reported to be under the control of JAK/STAT (see above).

### 4.5. Wound-Healing, NO and ABA in Honey Bees

Regarding the wound-healing/encapsulation response in honey bees, previous reports show that a wound injury induces the proliferation/differentiation of granulocytes, which at the same time shows its response through an increased production of NO [74]. The over-proliferation of certain kinds of hemocytes, which at the same time are the ones producing NO, could be indicative of JAK/STAT participation in the wound healing response of *A. mellifera*. As it was mentioned before, ABA is considered as a cytokine in animals’ immunity, and this kind of molecule is involved in the activation of the JAK/STAT pathway (see above). ABA has also been described to participate in honey bees’ immune responses, and specifically, an enhanced wound healing response was reported in ABA-fed honey bee larvae which had been previously challenged with *Varroa* [107]. In a subsequent study, the wound healing response, including the over proliferation/differentiation of granulocytes, was found to be enhanced by dietary ABA supplementation [75]. Interestingly, the over proliferation of hemocytes has been related with the action of the JAK/STAT pathway (please see this above in Section 4.1.). However, specific experiments should be performed to better describe the possible roles of NO and ABA during the wound healing response in honey bees.

## 5. *Varroa*-DWV Within the Context of Innate Immune Responses

### 5.1. The Importance of the Hemocytes’ Role

*Varroa destructor* and the associated DWV are a major threat to the world’s honeybees [146]. The active role of *V. destructor* in the dispersal and enhanced replication of the virus, related with the parasite’s feeding, indicates that the mite–virus association has clear benefits for the latter, and an adaptive value for the mite is also proposed [15,147]. Indeed, the delicate immune balance underpinning the covert infections of DWV can be destabilized by *Varroa* feeding, resulting in intense viral proliferation [146,148].

The feeding behavior of *V. destructor* is complex: The reproductive phase of the mite’s life cycle is characterized by a perforation of the abdominal sternites of the bee pupa by the mother mite. Through this open wound both the mother and its offspring repeatedly feed on the bee’s hemolymph [143,149,150,151]. Recently, Ramsey et al. [151] reported that *Varroa* also feeds from the fat body tissue of adult bees. The wound produced by the mouthparts of the mite should elicit a humoral and/or cellular immune response, including hemolymph clotting, melanization, or encapsulation, which would interfere with the parasite’s food uptake, and may result in reduced mite’s fitness. Thus, it is understandable that *Varroa* secretes anticoagulant factors into the mite’s saliva to counteract those responses [143]. In addition, the melanization and encapsulation responses are negatively correlated with DWV titer [16,81] and a significant negative correlation between DWV and immune gene expression (including *PPOact*) is reported [14,143,145,148].

In other insect species it has been extensively demonstrated that the activation of melanogenesis and encapsulation is directly linked to the immune activation of insects’ hemocytes [152,153]. These humoro-cellular responses are mediated by a number of genes that control the formation of a cellular capsule around foreign intruders and the deposition of melanin and other toxic molecules on their surface like ROS and RNS [62,126]. PO is initially synthesized by the hemocytes and released into hemolymph as inactive proPO, which is activated by a serine protease cascade upon the recognition of foreign invaders [153]. Interestingly, Ling and Yu [154] describe that the activation of proPO in the surface of *Manduca sexta* hemocytes may initiate melanin synthesis, leading to the systemic melanization of hemocyte capsules.

It is proposed that in honey bees, and more generally in insects, inducible antiviral barriers besides RNAi-mediated mechanisms may play an important role [16]. A recent study by Annoscia et al. [81] shows that mite feeding destabilizes viral immune control through the removal of both virus and immune effectors, triggering uncontrolled viral replication. But, by contrast, they do not find consistent support for alternative proposed mechanisms of viral expansion via mite immune suppression or within-host viral evolution.

Their results suggest that hemolymph removal, observed in the presence of feeding *Varroa* mites, plays a key role in the enhanced pathogen virulence, probably as a result of the reduction of the bee’s hemocytes. The latter is in great concordance with the relevance of hemocytes in the insect’s immunity (see above), particularly associated with the activation of the melanization response involved in wound-healing/encapsulation processes, which are critical to respond to parasites like *Varroa*, and had been linked to the JAK/STAT pathway (see above).

Possibly, as it is recently evidenced by Annoscia et al. [81], and previously proposed by Negri et al. [77], hemocytes could play a key part in defending the host, highlighting the relevance of cellular immunity in honey bees (Figure 1).

### 5.2. Varroa-DWV and Immune Response: Pathways Shared by JAK/STAT, Toll, ABA and NO Signaling

The impact of seasonal cold stress in association with high *V. destructor* infestation levels is of great concern from a practical and scientific perspective. The overwintering colony must have a large-enough bee population and food resources to support thermal stability. At the same time, the developing brood and adult bees should be able to deploy immune defenses to counteract physical injury and viral infections associated with the mite’s feeding.

The activation of the enzymatic cascade leading to melanization has been reported to be down-regulated by the *Varroa*-DWV association [144,145,148] in correlation with overwintering [14]. Interestingly, ABA-supplemented larvae were found to be able to revert the anticoagulant effects of *Varroa,* showing an improved cicatrization leading to a total closure of a wound injury [108]. Recent results show that ABA supplementation induces the up-regulation of the *PPOact* gene in *A. mellifera* larvae reared in vitro under standard temperature [109]. The effects of dietary ABA supplementation are also related to an augmented melanization response in adult bees, evidenced by an enhanced PO activity [107]. However, when ABA supplemented larvae are also submitted to cold stress, the up-regulation of *PPOact* is not observed [109].

*Varroa* secretes virus and anticoagulant factors into the bee larva’s blood to impair humoral and cellular immune responses related with wound-healing and encapsulation, such as melanization and hemocyte spreading [14,16,144]. Recently, it was demonstrated that *V. destructor* parasitism also affects honey bees through the remotion of hemocytes [81]. In the same work, the authors show that, as a consequence of that remotion, the encapsulation response is significantly diminished in *Varroa*-infested bees. Interestingly, in collapsing colonies affected by the *Varroa*-DWV association, the barrier under JAK-STAT control appear to be targeted [16] (Figure 1).

The immunosuppression related with viral replication is also caused by an upstream alteration of the Toll pathway [16]. Recently, Zhao et al. [155] reported that low DWV titers at early time-points coincide with high levels of the Toll pathway transcription factor *Dorsal*, a gene encoding a protein in the NF-κB family. Interestingly, when the viral titers of the bees increases, the levels of *Dorsal* decreases [155]. In concordance, Annoscia et al. [81] show a significant decrease in the expression of *Dorsal 1A* in bees with high levels of DWV. These results provide additional evidence for the active immune suppression by the DWV associated with the Toll pathway. Recently, Quintana et al. [156] found that bees with deformed wings and increased levels of DWV have upregulated expression of the genes *Domeless* and *TEPa* (JAK-STAT pathway), and also *NOS*. Interestingly, in the same study, no differences in *Toll Wheeler (Toll18W)* mRNA levels are found, but, an up-regulation of the NF-κB inhibitor (IκB) gene *Cactus* is detected in bees with deformed wings and increased levels of DWV. Together, the results described above suggest that the JAK-STAT pathway could be activated while the Toll pathway is inactivated in relation with *Varroa* infestation and high levels of DWV (Figure 1).

As it was previously described in this study, NO activates NF/κB-dependent cascades [126], is involved in response to viral infection [69,70] and is produced in honey bees’ hemocytes, participating in wound-healing responses [74]. Possibly, there could be a relationship between the level of *Varroa* parasitism (in time and/or number of mites), viral titers, the amount of NO-producing hemocytes, NF-κB signaling and the balance of the overall immune response of *A. mellifera* to *V. destructor*-virus association (Figure 1). However, this should be considered as a hypothesis, and specific experiments should be performed to test it.

Recently, Negri et al. [109] found that both the *Toll18W* and *Cactus* are induced by cold stress in *A. mellifera* larvae. Interestingly, in the same study, the expression of *Toll18W* is both induced in ABA-supplemented larvae reared in standard temperature conditions and also under cold stress. In both cases, when larvae are fed with ABA, the expression of *Cactus* is similar to the control situation (normal diet under standard rearing temperature). The results described above suggest that the Toll pathway is activated in response to low temperatures and ABA per se (Figure 1).

In the same study described before, both *Domeless* and *TEPa* from the JAK/STAT pathway are induced by cold stress in *A. mellifera* larvae [109]. In addition, *NOS* expression is both induced by cold stress and ABA supplementation alone, meaning that NO signaling could play a role in response to cold, and that ABA per se could induce NO synthesis in honey bees. At the same time, ABA does not induce the expression of *Domeless* or *TEPa*. Previously in this work, we also described that ABA augments the levels of NO production in hemocytes in honey bee larvae [75]. Here, we also reviewed that NO plays a crucial role signaling the activation of the JAK-STAT pathway and mediates the transmission of both intra- and intercellular signals in different model organisms. Thus, a possible connection between ABA and the JAK-STAT pathway could be suspected through the action of NO in *A. mellifera* (Figure 1).

It is important to highlight that in both works from Negri et al. [75,109] the experiments were performed using in vitro reared larvae. This is relevant to this analysis, because rearing bee larvae in vitro reduces the chances of having significantly different levels of DWV between the sampled individuals. Gregorc et al. [157] find augmented levels of DWV within in vitro reared larvae only when they artificially parasitize the larvae with *V. destructor*. In another work, Ryavob et al. [158] find augmented levels of sack brood virus (SBV) and DWV, and a response to those viruses, this in in vitro reared larvae only when they feed the larvae with the viruses. Together, the results described above seem to indicate that in standard conditions or larval rearing, the levels of SBV and/or DWV should not be significant.

As was mentioned in the introduction, bees’ immune suppression resulting from pesticides is an important factor that could seriously synergize with biotic stressors like *V. destructor* [159,160]. The phagocytosis innate immune response is activated through both the Toll and JAK/STAT pathways [59,161]. In 2013, Di Prisco et al. [162] found that clothianidin, a neonicotinoid insecticide, negatively affects NF-κB signaling in the Toll pathway of both flies and honey bees. Recently, Walderdorff et al. [79] demonstrated that the neonicotinoid imidacloprid (IMI) affects the immunocompentence of honey bees. In that work, they show evidences suggesting the interaction between IMI, LPS challenge, phagocytosis, H_2_O_2_ (a ROS, see above in the immunity section) and NO production in *A. mellifera* hemocytes. In 2017, Negri and co-workers [75] showed that hemocytes from LPS-injected *A. mellifera* larvae produce increased amounts of NO. In the same work, the authors show that when the larvae are fed with L-arginine (the natural substrate for NO production through NOS) the levels of NO in response to LPS increases. In some way, this last evidence connects NO with nutrition, because L-arginine has been reported to be an essential amino acid for bees [100].

Arefin et al. [163] find that the consumption of L-arginine enhances the melanization response in flies through NO production. These results are in concordance with previous data demonstrating that adding L-arginine to the food of *D. melanogaster* increases the ability of larvae to encapsulate the eggs of the parasitoid *Asobara tabida* [164]. In addition, Sanzhaeba et al. [165] demonstrate a dual effect of NO on the PO-mediated DOPA oxidation process of the melanogénesis process. A dual dose-dependent effect of NO on melanin generation is also demonstrated for isolated hemocytes of *Galleria mellonella* larvae [166]. The authors propose that the effects of NO take place in vivo, and that enhancement of melanization by NO may occur upon the encapsulation response within insects’ blood.

The activation of melanogenesis is triggered in response to wounding and encapsulation, and it depends on the immune activation of insects’ hemocytes [152,153]. It is reported that the activation of honey bee hemocytes is dependent of NO [73], and that *A. mellifera* granulocytes generate increased amounts of NO in response to wounding [74].

Interestingly, the effect of NO seems to be hemocyte-specific both in *D. melanogaster*, where NO production is related with lamellocytes [163,164] and *A. mellifera*, where this radical is shown to be generated in granulocytes [73,74,75]. Taking into consideration the results described in the two previous paragraphs, it could be speculated that NO could also play a (direct or indirect) role in melanogenesis in *A. mellifera*.

The results discussed above suggest that NO plays a key role in the innate immunity response of bees and other insect species. NO participates in responses from hemocyte’s activation (spreading) to melanization, contributing in encapsulation and wound healing, which are critical during *V. destructor* parasitism (Figure 1). The possible cytokine-like role of ABA and the interaction with NO and the pathways and responses described above should be further investigated (Figure 1).

### 5.3. Varroa-DWV and Environmental Factors: The Interaction Between the Immune System and Cold Temperatures

The concept of host regulation involves a wide range of host physiological and behavioral changes induced by parasites which are especially investigated in insect parasitoids [16]. Typical models of study are the parasitic wasps, which are able to colonize and exploit living insect hosts, using a wealthy repertoire of virulence factors [143]. The *Varroa*–DWV association could be considered a similar system where the vector role of *Varroa* is paid back by a DWV-induced fitness enhancement mediated by host immunosuppression, but at an earlier stage and with a less intimate level of integration [16]. A series of correlative studies also provides evidence supporting a major role of DWV in the immune suppression process, characterized by a negative impact on a member of the NF-κB protein family [16,81].

Recent reports suggest that winter cold weakens bee colonies by decreasing the expression of immune genes [14,17]. Steinmann et al. [14] hypothesize that while detrimental effects of *V. destructor* infestation (promoting DWV replication) occur during the whole life cycle of honey bees, its effect becomes critical in fall and winter when increasing mite infestation levels are concomitant with a seasonal decline in immune function, and the expected extended longevity of fall and winter bees. These authors support the hypothesis that fall represents a critical period when honey bee colonies experience important nutritionally-dependent, physiological adaptations to survive winter, and that food stores and *V. destructor* infestation levels are key factors determining the destiny of the colonies (Figure 1). Indeed, a number of studies identify and connect the effects of cold, the parasitic mite *V. destructor* and vectored viral pathogens, particularly DWV, in contributing to significant changes in the global viral landscape and a continuing decline in honey bee health [7,9].

If *A. mellifera* brood susceptibility to *V. destructor* infestation and other diseases is linked to cold and virus-induced immunodeficiency, studying the bees’ immune responses to biological threats and environmental stressors should help to alleviate the losses of managed honey bees. Still, few studies have bridged the gap between individual bee immunity and colony level defenses, and related to these combined elements, to useful measurable outcomes for beekeeping operations [9].

In respect with the discussion above, ecological immunology reveals an increasing web of relationships between immune responses, behavior and stress in a wide range of organisms [167]. In this context, it is important to consider that due to insects’ immune responses being energetically costly, they can compromise fitness [168] or be compromised by environmental stressors [169]. This last highlights the relevance of taking into consideration the integration of environmental factors that could be related with the time of the year associated with a particular disease affecting bees.

Insects are proposed to be the most successful group of terrestrial animals, having overcome the challenge of thermal variability [170]. In combination with this abiotic stress, there are biotic stresses like a diverse set of parasites and pathogens that affect the performance of insects’ immune and anti-stress responses [170]. Thus, overwintering temperate insects are useful models for understanding the interactive nature of multiple abiotic and biotic stressors, because cold stress during overwintering is frequently related with a trade-off with energy-consuming processes like immune responses [170].

Honey bees in temperate regions fit perfectly within the scenario described above.

However, the interactions between the immunity and environmental stress are complex and, interestingly, not necessarily negative [167]. Main components of the insect immune system like phagocytosis and encapsulation still operate (albeit slowly) at low temperatures [171]. Besides, insects exposed to cold show increased tolerance to fungal infection and an upregulated expression of immune-related genes [170]. This cold-associated upregulation may have ecological relevance; for example, a better encapsulation response is associated with enhanced winter survival in water striders [139]. Conversely, bacterial infection increases the time taken to recover from chill coma (i.e., reduces cold tolerance) in *D. melanogaster* [172]. Thus, there appear to be links between the responses to cold and infection and the nature of those responses and their adaptive significance represents an amazing field to explore in depth. This last paragraph leads us to the idea involving the possible cross-talk between shared routes in response to different stressors and the concept around “immune priming”, which is discussed below.

## 6. Key Molecules and Priming Effects

The primary immune responses of insects are innate, including both humoral and cellular defenses against parasites and pathogens. However, there are interesting evidences of the priming of the insect immune system by prior exposure to pathogens [173,174] or environmental stressors [175]. Interestingly, Le Bourg et al. [175] find that flies subjected to cold survive longer against fungal infection, while other stresses have no positive effect. This effect appears even when the cold pre-treatment is applied to young-aged flies, increasing their survival to infection throughout life [175]. However, we could also interpret immune priming beyond the boundaries of individual immunity, taking into consideration a superorganism as a model of study, in which overall immunity is built upon the integration of individual and social responses. In the case of honey bees, Richard and co-workers [85,176] demonstrate that individual immunity (stimulated both with bacterial elicitors or bacterial injection) triggers the social response of allogrooming, which is a social behavior associated with *Varroa* parasitization. Interestingly, these authors demonstrate that both the JAK/STAT and Toll pathways are significantly regulated in response to bacterial challenge. It could be hypothesized that social immunity could be induced (or primed) through the stimulation immune defenses at the individual level, thus enhancing the overall immune response of the superorganism, and that the JAK/STAT and Toll pathways should be a possible target to achieve those effects (Figure 1).

In this sense, we hypothesize that ABA treatment could be exerting a priming effect over the response to low temperatures, allowing honey bee larvae to be better prepared to confront cold stress or any related immune challenge sharing the JAK/STAT and Toll pathways and NO production as a common response (like for example, wound healing or virus infection) (Figure 1). At least in part, the hypothesis proposed in this work could explain the results obtained in field experiments where ABA-supplemented colonies [107] show an improved fitness, evidenced by the level of winter bees’ population, to pass through the winter season (Figure 1). However, more experiments should be performed to evaluate the possible relationship between ABA, bees’ physiology, immunity and *Varroa*-DWV parasitism. Future experiments should also evaluate the role of ABA integrating individual and social defenses, from molecular assays to field tests.

A schematic summary of everything analyzed, integrated and proposed in this review is shown in Figure 1.

## 7. Conclusions and Perspectives

The question of domestication, genetic diversity and colony survival is more relevant in recent times as researchers and honey bee breeders search for evidence of increased resistance to pathogens and parasites. Youngsteadt et al. [177] show that managed honey bee colonies have lower levels of immunocompetence than feral ones exposed to natural selective pressures, and this resistance appears to be linked to a higher genetic diversity [178]. These results highlight the high impact of anthropogenic selection on *A. mellifera*, and suggest that impaired immunocompetence acts as a key weakness in the bees’ capacity to face the biotic stress factors occurring in the hive ecosystem [7].

We need to search for natural solutions to respond to short and long term threats to honey bee health. In the short term, the goal is to reduce the rapid rate of colony loss, and hopefully try to stop it. In the long term, we should surround bees with favorable floral environments, and human actions must be in tune with the environment, its biology and its ecology.

If indeed nutrition can be used to maintain and/or elevate the immune defenses of bees, then the long term strategy must involve the preservation of natural environments and an increase in the availability of diverse floral resources in agro-productive ecosystems (Figure 1). This means including the best selection of flowers for bees, considering the floral design in relation with the seasons and crop production schemes. In the short term, one possibility is to supplement the diet with key molecules (like phytochemicals) found in the natural diet and physiology of bees (Figure 1).

Providing key molecules is different from standard supplemental feeding. The criteria around focusing on key molecules is related with achieving the desired effect via the addition of a small amount of the chemical within the bees’ diet. To be effective, these key compounds must participate in important signaling pathways for the immunological defense of bees. For that it is important to understand the signaling pathways and their relationship with the different stressors that are affecting the bees. Also, the levels and concentrations of any possible key molecule should be analyzed in natural foods sources as well as in the bee’s physiology. At the same time, it is important to take into account that, as mentioned above, social defenses could be triggered by the activation of immunity at the individual level. For that, it could be interesting to analyze the relationship between key molecules and the interaction of signaling pathways activation at individual and social immunity levels. This last would give us information regarding the roles of different molecules at the superorganism level, and that is a goal which any bee scientist should pursue.

As it was mentioned previously, immune pathways engage in cross-talk and can direct some of the same immune effectors [59]. This means that we could study which are the factors (biotic or biotic signals or stresses) that trigger each immune pathway, and if there is any cross-talk between the different signaling cascades. Then, understanding how different pathways could be induced, we could search for “priming” molecules, which hopefully could activate more than one response at the same time (through cross-talk).

The signaling pathways have well-defined routes and interactions sets, which can be examined to understand the resulting cascade of events, triggered by certain factors, related with each pathway. This kind of detailed examination could lead us to detect which signaling molecules are those responsible to connect the signal with the transduction in each pathway, which in turn could provide clues to what we call “key molecules”. In a sense, we could also take advantage of the cross-talk between different pathways to search for shared signals that boost those pathways or turn one “on”, while also switching the other “off”.

The last could lead to one of the keys to find what we call “key molecules”. Then we could select different combinations of “key molecules” to design, precisely, the diet of our bees. This could be achieved through adding the molecule itself, or adding the necessary molecules to induce the synthesis of a particular signal, etc.

In this way, we could develop what we call “Precision Nutrition”, within the framework of precision agriculture, and particularly, of precision beekeeping.

## Figures and Tables

**Figure 1 insects-10-00401-f001:**
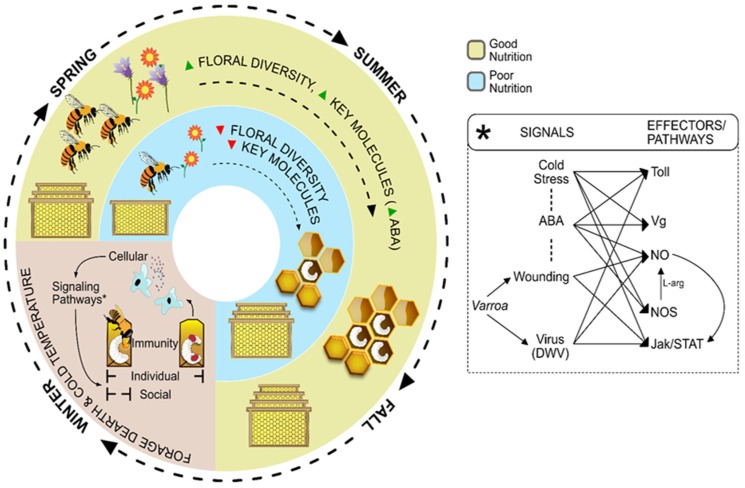
Schematic summary illustrating the relevance of nutrition during spring, summer and fall for colony growth before entering winter, which is characterized by forage dearth and cold temperatures. A good nutrition (green area of the diagram) is based upon the great diversity of floral resources and the availability of key molecules, like for example abscisic acid (ABA). In contrast, the lack of floral diversity combined with reduced amounts of key molecules leads to a poor nutrition (blue). Differences in nutrition lead to dissimilar levels of colony growth (represented visually in the figure as the different number of honeycombs). These colony-level variations in bee population lead to different capabilities to mount a socially-based immune response to confront winter-associated stresses. During winter, the immune strength of the colony is based in the interaction between social and individual immunity. Once any or all of these stresses escape the social level of defenses, the signal is captured, transduced and amplified by activated hemocytes (cellular) through the main signaling pathways reviewed here in association with Varroa, virus (deformed wing virus, DWV) and cold (Janus kinases/signal transducer and activator of transcription proteins (JAK/STAT) and Toll). The interaction between signals and effector pathways is illustrated in the right side of the main figure. In this model, nitric oxide (NO) represents a key molecule playing a role within the cross-talk between the responses triggered by *Varroa* (viruses and wounding) and cold. At the same time, ABA is proposed as a cytokine playing a priming effect over those responses, exerting its role through NO production activation. At the start of the foraging season (spring in the diagram), the post winter colonies may also differ in the number of forager bees able to start the cycle again when the blooming periods arrive.

**Table 1 insects-10-00401-t001:** Summary of the evidences found in literature related with phytochemicals and bees’ medication.

Evidences	Phytochemical	ABA	CouA	Quer	Nic	Ana	Caff
Level of organization	Molecular	[52,107,109]	[101]	[110,111]	[112]	[112]	[113,114]
	Cellular	[75,107]					
	Individual (organism)	[52,107]	[110,111,114]	[110,111]	[112,115,116]	[112,116]	[113,114]
	Colony (superorganism)	[107,117]					
Experimental system	In-vitro rearing	[52,75,109]				[112]	
	Laboratory	[111]	[101,110,111,114]	[110,111]	[112,115,116]	[112,116]	[113,114]
	Field	[107,117]					
Ontogeny	Larva	[52,75,107,109,117]					
	Pupa	[52,117]					
	Adult	[52,107,117]	[101,110,111,114]	[110,111]	[112,115,116]	[112,116]	[113,114]
Related with	Immune response	[75,107,109]			[112]		
	Parasite/Pathogen	[107,117]	[114]		[112,115,116]	[112,116]	[113,114]
	Pesticides	[107]	[101,110,111]	[110,111]			
	Abiotic stressors (not pesticides)	[52,109]					
Total references		5	4	2	3	2	2

ABA: Abscisic acid; CouA: P-Coumaric acid; Quer: Quercetin; Nic: Nicotine; Ana: Anabasine; Caff: Caffeine. Numbers within square brackets [] represent the citations found in the text and in the references list.

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
