# Peer review of "Towards Precision Nutrition: A Novel Concept Linking Phytochemicals, Immune Response and Honey Bee Health"

_insects, 2019, doi:10.3390/insects10110401_

Round 1

Reviewer 1 Report

This paper reviews what we know about honey bee immunity and how it can be influenced by nutrition. The authors hone in on the importance of hemocytes and abscicic acid. This is an excellent synthesis of the literature and is a worthy addition to our knowledge.

I encourage the authors to have somone with excellent knowledge of the English language edit the paper. At times, the grammar is perfect, at other times sentences and phrases are poorly written. 

The article is well organized, but there are a few topics that seem tangential. For example, with the focus of the paper on nutrition and immunity, sections 2.1.1, 2.1.4 and 2.1.5 seem to be asides. 

The title should be rewritten. Specifically, the current mention of "colony fitness" is inappropriate.  

Lines 109-110 are misleading, as the cited paper is about royal jelly, but the phrase indicates that the addition of honey and pollen to the larval diet has an effect. The original paper does not discuss honey and pollen.

I really appreciate Figure 1. It is an excellent summary.

I recommend this paper for publication after minor revisions.

Author Response

Reviewers' comments:

Reviewer 1:

Comments and Suggestions for Authors

First, we would like to thank the reviewers. The suggestions and comments made by them helped to improve the quality of the MS. We hope the reviewers to be happy with the revised version of the MS.

Our answers to each reviewer’s comment are written in italics.

This paper reviews what we know about honey bee immunity and how it can be influenced by nutrition. The authors hone in on the importance of hemocytes and abscicic acid. This is an excellent synthesis of the literature and is a worthy addition to our knowledge.

This comment is in concordance with the aim of the manuscript. We are happy for the reviewer´s apreciation of our work.

I encourage the authors to have somone with excellent knowledge of the English language edit the paper. At times, the grammar is perfect, at other times sentences and phrases are poorly written. 

Thank you very much for the comment. We are sorry for enlgish. The entire manuscript has been entirely revised to obtain an improved version of the language in the resubmussion.

The article is well organized, but there are a few topics that seem tangential. For example, with the focus of the paper on nutrition and immunity, sections 2.1.1, 2.1.4 and 2.1.5 seem to be asides. 

We understand the reviewer’s concern. We agree in that these sections may not represent the main focus of the manuscript. However, we think that these topics should be at least present in our work, because honey bees respond to different kind of factors individually and as a superorganism (socially) and that it is all integrated at the same time. For example, we need to introduce the topic “nest-hygiene” because later in the manuscript we explain how this social response is linked to individual immunity when bees involved in nest hygiene recognise infected bees. However, having in consideration the reviewer’s observation, we included phrases in the revised manuscript to highlight the connection of what is proposed above with the purpose of this review. We hope the reviewer to be satisfied with our response and to like the new version of the manuscript. 

The title should be rewritten. Specifically, the current mention of "colony fitness" is inappropriate.  

Accepted and changed. Please see the new version of the title. We hope the reviewer to be happy with the new title.

Lines 109-110 are misleading, as the cited paper is about royal jelly, but the phrase indicates that the addition of honey and pollen to the larval diet has an effect. The original paper does not discuss honey and pollen.

The reviewer is right. We apologise for the mistake. The sentence has been deleted because we also realized that it make no sense there. We hope the reviewer to be in agreement with the actual version of the paragraph.

I really appreciate Figure 1. It is an excellent summary.

Thank you very much for this comment. We really apreciate it because we putted a lot of effort to congregate all the idea into one clear figure.

I recommend this paper for publication after minor revisions.

We are happy and we thank the reviewer for the recomendation for publication after minor revisions. 

Reviewer 2 Report

The subject of this review is very interesting and innovative. However, it deserves to be reworked.

Indeed, the beginning concerning the descritption of bee immunity is well described and clear even if some results are cited as statements and should be more tempered, but later it becomes very difficult to follow.

Some parts like nutrition would ask for more details and be thorough, and the part on varroa / cold weather needs to be restructured with more subparts because the purpose of each paragraph is unclear and redundant.

Similarly I'm not sure, that the title is adapted. The aspect of the diet is only understood at the end of the article, and you do not really provide evidence for the choice of your key molecules but rather just interesting assumptions and perspectives, if I have well understood. So, it can not be used in the title according to me. It's a tool proposal. How could you bring ABA and NO to bees in the diet, did not I understand that?

Moreover, you could tweak the last two parts to clarify the purpose and the conclusions / perspectives.

line 107: the foragers also bring the mineral elements, the lipids, all the elements which need the bees.

107-108: Be careful it has not been proved that the bee selects the pollen for its nutritional quality unlike the drone or some wild bees.

On the other hand, there are differences in quality depending on the pollen collected. but the quality of the pollen is not enough, there is also the quantity and diversity. you talk about it below lines 253-254 but you take it for granted while it deserves to be described as the rest. quote any references. like the different works of singh. it is not enough just to bring diverse and abundant pollen; it is also necessary that the majority is of good quality. you can talk about the fact that diversity makes it possible to reduce the differences in quality or abundance found in the environment according to the season (see di pasquale et al., 2016).

In the thermoregualtion part, I would specify the temperatures of a "normal" hive from inside to outside. there is a bibliography that specifies the extreme temperatures felt as a stress depending on the species of bees, it would be worth to mention a few.

lines 156-157: I would temper this assertion because, as they say, aggression depends mainly on the genetics and environment of bees at a young age.

Author Response

Reviewers' comments:

Reviewer 2:

Comments and Suggestions for Authors

First, we would like to thank the reviewers. The suggestions and comments made by them helped to improve the quality of the MS. We hope the reviewers to be happy with the revised version of the MS.

Our answers to each reviewer’s comment are written in italics.

The subject of this review is very interesting and innovative. However, it deserves to be reworked.

We would like to thank the reviewer for the positive comment regarding our manuscript. At the same time, we agree in that the manuscript needed more work and we hope the reviewer to be happy with the revised version of it. We made a lot of rearrangements to improve the understanding of our proposal.

Indeed, the beginning concerning the descritption of bee immunity is well described and clear even if some results are cited as statements and should be more tempered, but later it becomes very difficult to follow.

We are happy to know that the reviewer think that the description of immunity is well described and clear. At the same time, we agree with the reviewer´s observation so the results have been tempered in the new version of the manuscript. We hope the reviewer to be happy with our work on that.

Some parts like nutrition would ask for more details and be thorough, and the part on varroa / cold weather needs to be restructured with more subparts because the purpose of each paragraph is unclear and redundant.

Accepted and changed. Please see the new version of the manuscript. We have reorganized the sections of the manuscript and added subparts within the longest sections. We hope the reviewer to like this new presentation of our work.

SimilarlyI'm not sure, that the title is adapted. The aspect of the diet is only understood at the end of the article, and you do not really provide evidence for the choice of your key molecules but rather just interesting assumptions and perspectives, if I have well understood. So, it can not be used in the title according to me.It's a tool proposal. How could you bring ABA and NO to bees in the diet, did not I understand that?

Please see the new version of the title. We hope the reviewer to be happy with the new title. Regarding the questions, we would like to make reference of the correlative series of works in which it has been previously demostrated that:

ABA can be added into the bees’ diet and this results in augmented levels of ABA in the treated individuals: In 2015, Negri et al. demonstrated, in a field experimetn, that ABA could be mixed with sugar syrup to suplement beehives and the the bees from thos beehives showed increased ammounts of ABA. In 2017, Ramirez et al. demonstrated the same but using an in-vitro-rearin technique, in laboratory conditions. In addition, ABA suplementation was performed in: Negri et al. 2017 and 2019 and in Szawarski et al. 2019 (this last has been recently accepted for publication within the same special issue of Insects). Please find the citations above in the references list of the manuscript. Regarding NO: This mollecule is produced, primarily, through the action of NOS enzymes in animal cells using L-arginine as the substrate. The roles of NO in Apis mellifera immunity, described up to this moment, were reported in Negri et al. 2013; 2014; 2017 and 2019. Particularly, the report of Negri et al. 2017, describes that L-arginine could be added into the diet of in-vitro reared larvae and this resulted in augmented levels of NO in response to lipopolysacharide. Please find the citations above in the references list of the manuscript.

We would like to comment also, from author to reviewer, that we have non-published results involving L-arginine suplementation of bee hives. However, these results can´t be cited in this manuscript.

Regarding the choice of the key molecules:

The reasons for choosing ABA and NO as key molecules related with bee nutrition and immunity is because, through a series of correlative papers, these two mollecules gather the best amount of evidences at the moment. Evidences that has been demonstrated through different experimental approaches, at different levels of organization and in response to different kind of threats like:

At molecular level:

Results obtained in-vitro-reared bees (Ramirez et al., 2017; Negri et al., 2019).

Results obtained from bee samples from the field (Quintana et al., 2019)               

At cellular level (uni and multi cellular responses):

Results obtained from in-vitro-reared bees (Negri et al., 2017)

Results obtained from bees taken from colonies in the field and the experiments were performed at the lab (Negri et al., 2013; 2014; 2015a)

Results obtained from bees taken from colonies involved in a field assay (the experiment was conducted at field level) (Negri et al., 2015)

At individual level (organism):

Results obtained from bees taken from colonies involved in a field assay (the experiment was conducted at field level) (Negri et al., 2015)

At colony level (superorganism):

Results obtained in field, using small colonies as first step to search for effects at colony population (Negri et al., 2015).

Results obtained in field, using Langstroth colonies as second step to search for effects over adult and brood population (Negri et al. 2015; Szawarski et al. 2019, recently accepted for publication this same issue).

Overall, the series of papers cited above, studied the responses to the following challenges:

Non-self surfaces (recognition and activation at uni and multicellular level); bacterial elicitors; wounding; wounding recovery after Varroa parasitism; cold stress per-se; colony overwintering; Nosema Szawarski et al. 2019 (recently accepted for publication this same issue).

All the articles cited above are included within the text of the new version of the manuscript and the reference list.

However, taking into consideration the reviewer’s observation, the reasons to focus on ABA as phytochemical were summarized in the table 1, included in the revised version of the manuscript. This table also compares the amount of research in relation with the major phytochemicals reported. Thank you very much for motivating us to make this proposal clearer.

Moreover, you could tweak the last two parts to clarify the purpose and the conclusions / perspectives.

We reorganized the manuscript. Please see the new version of it. We hope the reviewer to be pleased with the revised version of the manuscript.

line 107: the foragers also bring the mineral elements, the lipids, all the elements which need the bees.

We agree with the reviewer´s observation. We include a phrase in relation with the comment. We hope the reviewer to be satisfied with the new version of the phrase.

107-108: Be careful it has not been proved that the bee selects the pollen for its nutritional quality unlike the drone or some wild bees.

The reviewer is wright. We apologise for that and we thank for the observation. We corrected this in the revised manuscript. We hope the reviewer to be pleased with the actual sense of the revised version of the sentence.

On the other hand, there are differences in quality depending on the pollen collected. but the quality of the pollen is not enough, there is also the quantity and diversity. you talk about it below lines 253-254 but you take it for granted while it deserves to be described as the rest. quote any references. like the different works of singh. it is not enough just to bring diverse and abundant pollen; it is also necessary that the majority is of good quality. you can talk about the fact that diversity makes it possible to reduce the differences in quality or abundance found in the environment according to the season (see di pasquale et al., 2016).

We would like to thank the reviewer for the proposed article (di Pasquale et al 2016). The main findings described in that work were introduced in the revised version of the manuscript. At the same time, some changes were made within that section related with pollen (quantity, quality and diversity) and immunity. In addition, we would like to propose the reviewer to consider the findings described by Kriesell et al. We hope the reviewer to be in agreement with the new version and content of those paragraphs.

In the thermoregualtion part, I would specify the temperatures of a "normal" hive from inside to outside. there is a bibliography that specifies the extreme temperatures felt as a stress depending on the species of bees, it would be worth to mention a few.

We agree with the reviewer`s comment. We have modified the paragraph within those lines and included two more relevant citations. However, although thermoregulation is an interesting theme, we didn’t want to extend too much this section. We hope the reviewer to understand this and, at the same time, to be satisfied with the modifications made within the manuscript.

lines 156-157: I would temper this assertion because, as they say, aggression depends mainly on the genetics and environment of bees at a young age.

We agree with the reviewer. The assertions between those lines of the original manuscript have been tempered in the revised version.

Round 2

Reviewer 2 Report

I thank the authors for considering our comments, and for making many changes to the article that make it more complete and clearer.

Indeed, the different parameters and the links between them (immunity / nutrition / winter) are better described and clearer. However, I think that there is a lack of linkages between the parts that help to conclude each part and to introduce the next one according to the conclusions obtained and the question for the next part (especially as you do not have final conclusions but rather perspectives). You did it very well, for example, at the beginning of part 4.2. Indeed, it helps to better understand where we are, and titles and subtitles help a lot. But the article is long, so very complete, but we get lost a little sometimes, or it may seem sometimes just be a list of examples one behind the other unrelated. Although it should be noted that great efforts have been made in this direction.

I confirm that I appreciate the review part of the bee immunity as few are so complete. Many publications are cited, old but also many recent, which seem important for the areas studied.

In the “2.1.5. part”,  as you say, aggression is a parameter that is very variable and depends on many parameters (species, environment, stressors), but it is primarily defined by genetics, hence the observed differences in hive to another of the same species in the same apiary. It just deserves to be mentioned among the cited factors.

I find the “3.2. part’” added very interesting. In the sentence: "Interestingly, when couAwas added to the diet of the midgut metabolism of the acaricide coumaphos in the midgut increased significantly", the "midgut" is not pronounced once too much?

Attention, I do not know if it's only me or the version, but I could not see the last line of Table 1 because the legend covered it.

Personally, despite your great efforts to clarify things, I still find the explanation on the JAK / STAT pathway difficult to understand. Maybe adding a schema could help. What do you think about it?

Figure 1 is not clear alone, but the legend is very well explained fortunately.

I really like your perspective part which now makes more sense.

Author Response

Mar del Plata, 1 of November 2019

Dr. Silvio Erler, Dr. Simon Tragust
Assigned Editors of special issue "Biology of Social Insect Diseases"

Insects

Dear Editors,

We have submitted the revised version (R3) of the manuscript entitled “Towards precision nutrition, a novel concept linking phytochemicals, immune response and honey bee health " (Manuscript ID: insects-573023-R3) co-authored with Ethel Villalobos, Nicolás Szawarski, Natalia Damiani, Liesel Gende, Melisa Garrido, Matías Maggi, Silvina Quintana, Lorenzo Lamattina and Martin Eguaras.

We are very grateful with the positive response and considerations of the editors and the reviewer about the revised version (R2) of the manuscript. In this new version, we have addressed all the minor comments and suggestions made by the reviewer.

Below we provide a point-by-point list of responses with our answers in italics. We hope the editors and the reviewer will be satisfied with the corrections made to the manuscript and with our responses to the reviewer’s concerns. We also hope the manuscript to be in an acceptable form for publication in Insects.

We would like to express our thanks for your editing efforts to our manuscript and look forward to hearing from you soon.

Yours sincerely,

Dr. Pedro Negri

Centro de Investigación en Abejas Sociales (CIAS)

Universidad Nacional de Mar del Plata

Consejo Nacional de Investigaciones Científicas y Técnicas (CONICET)

Argentina

Point-by-point list of responses to reviewer's comments:

a. I thank the authors for considering our comments, and for making many changes to the article that make it more complete and clearer.

First, we would like to thank the reviewer for his work. Then we would like to express that we are thrilled to know that the changes made the manuscript more complete and clearer.

b. Indeed, the different parameters and the links between them (immunity / nutrition / winter) are better described and clearer.

Thank you, we are pleased for that.

c. However, I think that there is a lack of linkages between the parts that help to conclude each part and to introduce the next one according to the conclusions obtained and the question for the next part (especially as you do not have final conclusions but rather perspectives).

We agree with the reviewer so we tried to improve the connectivity between each part of the manuscript. We hope the reviewer to be satisfied with our work now.

d. You did it very well, for example, at the beginning of part 4.2. Indeed, it helps to better understand where we are, and titles and subtitles help a lot.

Thank you, we are happy to know that we did that well. Thank you very much for supporting with that expression because it encourage us to keep working and at the same time, it served as an example for us to improve the other section of the work.

e. But the article is long, so very complete, but we get lost a little sometimes, or it may seem sometimes just be a list of examples one behind the other unrelated. Although it should be noted that great efforts have been made in this direction.

We agree with the reviewer in that the article is long. We tried to shorten it in this new version. However, as the reviewer also expressed, we tried to make this review as complete an integrative as possible. At the same time, it is an analytical review, not just descriptive. It also results in a compromise situation because while increasing connectivity of the different sections we would be increasing the length of the manuscript. We made our best effort to respond to the reviewer’s concerns regarding the connectivity of the examples with the idea of the work, making the reading easier.

f. I confirm that I appreciate the review part of the bee immunity as few are so complete.

Thank you very much. We are very grateful for your appreciation

g. Many publications are cited, old but also many recent, which seem important for the areas studied.

We are pleased to know that the reviewer noticed and highlighted it. For us, it is very important to take into consideration the bases of some areas and integrate old and new reports to better understand biological processes. The same consideration could be done regarding integrating different biological systems, going from vertebrates to invertebrates and vice versa.

h. In the “2.1.5. part”,  as you say, aggression is a parameter that is very variable and depends on many parameters (species, environment, stressors), but it is primarily defined by genetics, hence the observed differences in hive to another of the same species in the same apiary. It just deserves to be mentioned among the cited factors.

We agree. We made modifications in the text to clarify this. Please see the changes in the new version of the manuscript.

i. I find the “3.2. part’” added very interesting.

We are happy to know that the reviewer finds the section 3.2 very interesting. Thank you very much for sharing this. It is important for us to know that adding that section was a good decision.

j. In the sentence: "Interestingly, when couAwas added to the diet of the midgut metabolism of the acaricide coumaphos in the midgut increased significantly", the "midgut" is not pronounced once too much?

We agreed with the reviewer’s concerns about that phrase when we read it from her/his comment above. So, we looked for the sentence in the version of the manuscript uploaded by the editor to be revised for us. We were surprised to find out that the sentence was: “Interestingly, when CouA was added to the diet the metabolism of the acaricide coumaphos in the bees’ midgut increased significantly (table 1)”. In that sentence, the word midgut is present one time. However, we would like the reviewer to evaluate if she/he agrees in that the actual sentence is Ok.

k. Attention, I do not know if it's only me or the version, but I could not see the last line of Table 1 because the legend covered it.

We are sorry to know that and we apologize if we made a mistake in the edition of the table and its’ legend. In the version provided by the editor for us to revise, the last line is visible. Please check this in the present revised version of the manuscript. The last line of table 1 is referred to as “Total references”. Taking into consideration the differences between the reviewer’s observation here and the one above (referred to the repetition of the word “midgut”) with the version that we have (uploaded by the editor) and have read now, we suspect that there could be some mistakes in the version provided to the reviewed for revision. Maybe, troubles arise by the text processor used?...

l. Personally, despite your great efforts to clarify things, I still find the explanation on the JAK / STAT pathway difficult to understand. Maybe adding a schema could help. What do you think about it?

We understand the reviewer’s concern. We made our best effort to improve the narrative regarding the relevance of the JAK/STAT pathway in our proposal.

We agree that a scheme could be helpful, however, we are afraid about including one image referred specifically to the JAK/STAT pathway because it could distract the attention of the reader. The JAK/STAT pathway is analyzed in this manuscript and it is one of the two main pathways involved in the integrative analysis. The other pathway involved is the Toll. However, both pathways were used as an example of how we could integrate previously reported responses and pathways in different models species triggered by different biotic and abiotic stresses and how those stresses could be connected for sharing signaling pathways (like, for the present study, Toll). We think that, adding a specific scheme dedicated to the JAK/STAT could put too much focus only on this pathway. If the reviewer thinks that including a scheme is mandatory for this work, we could include one. However, we would like the reviewer to read the new version of the manuscript first and also see the scheme in figure 1 (on the right). There, the connection between different stressors, molecules, and the JAK/STAT pathway is illustrated with arrows (the same with Toll). We would like the reviewer to have this response into consideration. 

m. Figure 1 is not clear alone, but the legend is very well explained fortunately.

We are happy to know that the figure and the legend, combined, explain the idea. It took a great effort to summarize the general idea in one image. We think that it is good to have all integrated into the figure.

n. I really like your perspective part which now makes more sense.

We are very pleased to know that the reviewer like the perspectives part. Thank you.